# Good practices to optimise the performance of maternal and neonatal quality improvement teams: Results from a longitudinal qualitative evaluation in South Africa, before, and during COVID-19

Willem Odendaal[1,2,3]*, Mark Tomlinson[4,5], Ameena Goga[1,6], Yages Singh[1], Shuaib Kauchali[7], Carol Marshall[8], Yogan Pillay[9,10], Manala Makua[8,11], Terusha Chetty[1,12‡], Xanthe Hunt[4,13‡]

1 HIV and Other Infectious Diseases Research Unit, South African Medical Research Council, Cape Town, Western Cape, South Africa, 2 Department of Psychiatry, Stellenbosch University, Cape Town, Western Cape, South Africa, 3 Health Systems Research Unit, South African Medical Research Council, Cape Town, Western Cape, South Africa, 4 Institute for Life Course Health Research, Stellenbosch University, Cape Town, Western Cape, South Africa, 5 School of Nursing and Midwifery, Queens University, Belfast, United Kingdom, 6 Department of Paediatrics and Child Health, University of Pretoria, Pretoria, Gauteng, South Africa, 7 Division of Community Paediatrics, Department of Paediatrics and Child Health, University of the Witwatersrand, Johannesburg, Gauteng, South Africa, 8 South African National Department of Health, Pretoria, Gauteng, South Africa, 9 Clinton Health Access Initiative, Pretoria, Gauteng, South Africa (formerly), 10 Division of Public Health and Health Systems, Department of Global Health, Stellenbosch University, Cape Town, Western Cape, South Africa, 11 University of South Africa, Pretoria, Gauteng, South Africa, 12 Discipline of Public Health Medicine, School of Nursing and Public Health, University of KwaZulu-Natal, Durban, KwaZulu-Natal, South Africa, 13 Africa Health Research Institute, Somkhele, KwaZulu Natal, South Africa

‡ TC and XH are joint senior authors on this work.
* willem.odendaal@mrc.ac.za

## Abstract

Many maternal and neonatal deaths can be avoided if quality healthcare is provided. To this end, the South African National Department of Health implemented a quality improvement (QI) programme (2018–2022) to improve maternal and neonatal health services in 21 public health facilities. This study sought to identify good practices aimed at improving QI teams' performance by identifying optimal facility-level contextual factors and implementation processes. We purposively selected 14 facilities of the 21 facilities for a longitudinal qualitative process evaluation. We interviewed 17 team leaders, 47 members, and five QI advisors who provided technical support to the teams. The data were analysed using framework analysis. We choose the Consolidated Framework for Implementation Research as framework given that it explicates contexts and processes that shape programme implementation. Six quality improvement teams were assessed as well-performing, and eight as less well-performing. This research conceptualises a 'life course lens' for setting up and managing a QI team. We identified eight good practices, six related to implementation processes, and two contextual variables that will optimise team performance. The two most impactful practices to improve the performance of a QI team were (i) selecting healthcare workers with

**Data Availability Statement:** The full data set cannot be shared publicly because of the regulations imposed by the three respective Human Research Ethics Committee's (HRECs) who approved the study. Access to the data can be requested from the South African Medical Research Council, Human Research Ethics Committee, Chairperson, Prof. Danie du Toit. The request should be send to the Committee's chief administrative officer: Adri Labuschagne: Work phone number: +27 21 938 0687, Email address: adri.labuschagne@mrc.ac.za.

**Funding:** The evaluation was funded by ELMA Philanthropies (20-P0001-C), an anonymous donor, and the South African Medical Research Council. The funders did not play a role in designing the study, data collection and analysis, our decision to publish, and drafting the manuscript.

**Competing interests:** NO authors have competing interests The authors have declared that no competing interests exist.

quality improvement-specific characteristics, and (ii) appointing advisors whose interpersonal skills match their technical quality improvement competencies.

## Introduction

Globally, improving the quality of maternal and neonatal healthcare (MNH) services is imperative to reduce maternal and neonatal deaths and stillbirths, of which approximately 25% in low-and middle-income countries (LMICs) are preventable through improved pre- and postnatal care [1]. Quality improvement (QI) is premised on the idea that incremental, evidence-based changes to delivery processes lead to better outcomes [2]. A particularly common QI model, the Plan-Do-Study-Act (PDSA) model [3, 4] entails designated healthcare worker (HCW) teams selecting an area for improvement, developing (*Plan*), and executing (*Do*) the changes through iterative cycles. As the change (henceforth 'change idea') is implemented, its effectiveness is assessed (*Study*) [5]. The change idea involves modest changes within the HCW team's authority and existing resources [6, 7]. Based on the *Study* results, the team abandon, adopt, or modify the change idea (*Act*) [8].

QI can improve MNH services [9–11], yet, as concluded in a review of QI effectiveness assessed in randomised controlled trials, the evidence is mixed [12]. In South Africa (SA), combining the PDSA model with a learning network (when several QI teams meet to share their QI work) significantly improved HIV treatment initiation. [13]. Similarly, it significantly increased syphilis testing during antenatal care in Ethiopia [14], and in India, PDSA alone improved labour partograph completion [15]. However, in a Zimbabwean study, only two of the seven targeted MNH outcomes improved [16]. In Nigeria, there were no significant differences in retention of mothers living with HIV six months post-delivery, between the PDSA plus learning network intervention group and controls [17]. In another Indian study, PDSA alone had no effect on stillbirths and neonatal mortality [18].

While some of the variable success of QI models is due to the underlying complexity of the target service, it is mostly attributable to contexts and implementation processes such: as the resources in the healthcare facility [19, 20]; leader characteristics [21, 22]; how well the team works together [23, 24]; and technical support from the QI advisor (henceforth 'advisor') [16, 25].

In SA, there has been a reduction in maternal and neonatal mortality and stillbirths [26–28]. Though neonatal mortality decreased from 12 deaths per 1000 live births in 2012 [27] to 10.7/1000 in 2018 [29], up to 50% of these were preventable [30]. COVID-19 led to an estimated 4.8% increase in these mortalities [31]. To accelerate the reduction of MNH mortalities, the National Department of Health (NDoH) developed a multi-partnered (S1 Table) QI programme to improve MNH services in public health facilities [32]. The programme, *Mphatlalatsane* (meaning 'morning star'), was implemented in 21 facilities across three provinces between 2018 and 2022. *Mphatlalatsane* was a multi-component programme and delivered across all levels of the health system [32]. At the facility level, it focused on establishing QI teams recruited from the facilities' HCWs to implement the PDSA model [32].

The NDoH commissioned the South African Medical Research Council (SAMRC) to assess *Mphatlalatsane's* effectiveness to reduce maternal and neonatal deaths and stillbirths, improve mothers' experience of care, and strengthen MNH service indicators, for example, adhering to the protocol for treating pregnancy-induced hypertension [32, 33]. The evaluation included a qualitative process evaluation to understand contextual and implementation influences at the macro-level (national and provincial), meso-level (district and sub-district), and micro-level

(facility). The micro-level evaluation–with which this paper is concerned–has, to date, produced two publications describing the early implementation stages and how facility QI teams adapted the PDSA model [33, 34]. The macro and meso level evaluation was conducted by other members of the evaluation team and are reported elsewhere [35, 36]. The evaluation team were independent of the implementation team, and was not responsible for decision-making or implementation of the Mphatlalatsane programme.

To develop the findings of the micro-level evaluation into a conceptual contribution for QI programmes, this paper:

1. describes the characteristics of an optimal QI programme;

2. outlines the implementation contextual factors and processes over the life course of a QI team; and

3. suggests eight good practices to optimise team performance.

By synthesising the results through a 'life course lens', and expanding and refining good practices in relation to the Consolidated Framework for Implementation Research (CFIR) [37], we aimed to distil priority practices for future QI programmes in LMICs.

## Materials and methods

### Setting

*Mphatlalatsane* was implemented in the Mpumalanga, Limpopo, and Eastern Cape provinces (Fig 1), with seven facilities per province (see S2 Table for district data). Table 1 summarises the number of participating primary healthcare (PHC) clinics, community healthcare centres (CHCs), district hospitals, and regional hospitals (see S3 Table for details of the services provided at each facility type). NDoH ranked the facilities which provided birth delivery services using a provincial perinatal indicator matrix. The matrix included institutional maternal mortality ratios, neonatal mortality rates, and stillbirth rates based on the NDoH's District Health Information System [38]. NDoH selected facilities with the lowest scores on all indicators to participate in the *Mphatlalatsane* programme.

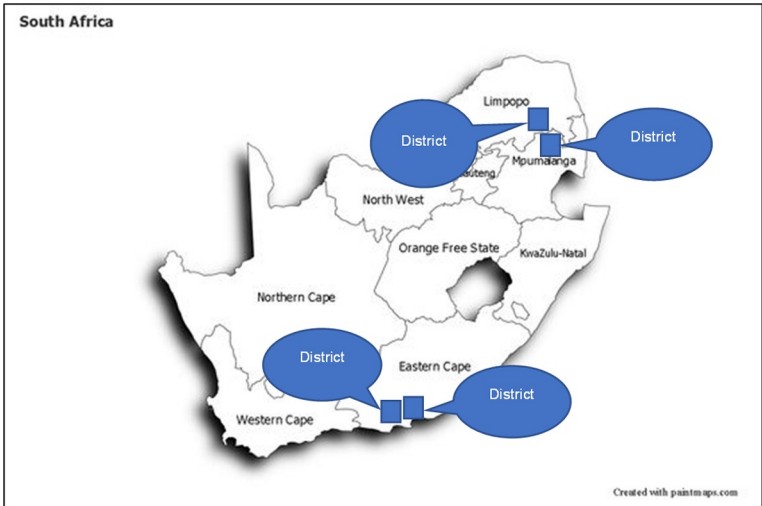

**Fig 1. Mphatlalatsane health districts.**

Table 1. All *Mphatlalatsane* and study facilities.

| Type of facility | *Mphatlalatsane* and Study facilities | District 1 | District 2 | District 3 | District 4 | Total |
|---|---|---|---|---|---|---|
| Regional hospitals | All *Mphatlalatsane* facilities | *1* | *1* | *1* | *-* | *3* |
| | Study facilities | 1 | 1* | - | - | 2 |
| District hospitals | All *Mphatlalatsane* facilities | *2* | *2* | *1* | *1* | *6* |
| | Study facilities | 2 | 1 | - | 1 | 4 |
| Community healthcare centres | All *Mphatlalatsane* facilities | *2* | *2* | *1* | *1* | *6* |
| | Study facilities | 2 | 2 | 1 | - | 5 |
| Primary healthcare clinics | All *Mphatlalatsane* facilities | *2* | *2* | *1* | *1* | *6* |
| | Study facilities | 1 | 1 | - | 1 | 3 |
| **Total** | **All *Mphatlalatsane* facilities** | *7* | *7* | *4* | *3* | *21* |
| | **Study facilities** | **6** | **5** | **1** | **2** | **14** |

*Facility with two teams

The NDoH embedded *Mphatlalatsane* within existing services by tasking the provincial departments of health and respective districts with managing the programme. HCWs within existing MNH service teams at the facilities were recruited as QI team leaders and team members (henceforth 'leaders' and 'members', respectively). They were to integrate QI activities into their regular work. The NDoH appointed the Institute for Healthcare Improvement (IHI) as the QI training partner. Facilities followed similar steps to establish the teams: management, consulting with senior staff, identified two to three staff for the IHS's PDSA training. The trainees and facility management then selected the leader from the trainees and recruited staff from existing service teams as members. Teams averaged between four to 12 members. IHI's involvement as the *Mphatlalatsane* QI partner ended around Month 11 of the project. For this reason, and the COVID-19 disruptions, the *Mphatlalatsane* QI model was not a fully-fledged IHI's QI collaborative network model [39].

The one component external to the existing systems was the advisors who provided mentoring and technical support to the teams. There was one advisor each in Districts 1 and 2, and one supporting both Districts 3 and 4.

Districts 1 and 2 teams were trained separately in September 2019, and Districts 3 and 4 teams jointly in February 2020. District 1 teams attended two more trainings in March 2020, before the COVID-19 lockdown was affected. The lockdown prevented further training for Districts 2–4. Whilst Districts 1 and 2 had six months of implementation and in-person advisor support, Districts 3 and 4 had only 3 weeks of implementation with advisor support before lockdown. A Programme management committee, comprising the *Mphatlalatsane* partners, oversaw its implementation. The NDoH invited the partners, who each had a dedicated role (S1 Table), for instance the Clinton Health Access Initiative managed the day-to-day activities, as well as appointing and managing the advisors. More detail on the preparatory work and programme management is described elsewhere [32, 33].

## Sampling

Due to budget constraints, 15 of the 21 facilities were purposively selected for this research, of which staff from 14 consented to participate. Staff in the facility who declined to participate felt that COVID-19 disrupted their team too severely for meaningful participation. In the 14 consenting facilities, there were a total of 15 teams: one per facility for 13 facilities, and one facility whose management opted to establish two teams (Table 1). A District 4 leader consented to her participation only, but declined the participation of their team, as they felt their

participation was sufficient. We ensured, in consultation with the advisors, that the selected facilities were a fair representation of facilities serving rural and urban communities, and well and poor scoring facilities on a *Mphatlalatsane* readiness assessment conducted by the NDoH before implementation commenced. Given their wider range of MNH services, more CHCs and hospitals were sampled than clinics.

## Participants

**Leaders.** Across the 14 participating facilities, 17 leaders, two of the fifteen teams had leader replacements during the evaluation, consented to participate, with one being a physician and the rest, midwives. Their median years as health professionals were 28 years, as managers, eight years, and 20 years at the facility, with District 4's leaders less experienced in all three characteristics (Table 2).

**Members.** The 47 member participants were recruited by their leaders. One member was a data clerk, and the rest were nurses. Districts 1–3 team members were more experienced and had been based at their facilities for longer periods, with medians of 23 and 19 years, respectively, versus the eight and three years for the District 4 team (Table 2).

**Advisors.** With two of the initial three appointed advisors resigning and being replaced, there were five advisors who all consented to partake in the evaluation. The Districts 3 and 4 advisor resigned in March 2021 and was replaced in August 2021, leaving these teams for four months without advisor support. The District 1 advisor resigned in July 2021 and was replaced in September 2021, leaving these teams for one month without advisor support. The District 2 advisor left in August 2022 and was not replaced. For the remaining five months of the programme the District 2 teams were supported by the District 1 advisor. As the evaluation team we were not informed as to why they resigned. We did not have contact with them post-resignation nor permission to add this question to our interview guide. The resignations appeared to be career moves and not related to their work as one enrolled for a PhD, and the other two for permanent positions elsewhere. Four advisors had substantive QI training with between 17 months and 15 years' advisor experience. The fifth advisor had much less QI training and experience.

## Data collection

The evaluation was conducted between February 2020 and November 2022 and consisted of the data collection methods summarised in Table 3.

**Table 2. Team leaders' and members' experience and years at the facility.**

| District | Number of facilities | Personnel category | Number of staff | Median years as health professional (interquartile range) | Median years at facility (interquartile range) | Median years as manager (interquartile range) |
|---|---|---|---|---|---|---|
| 1 | 6 | Leader | 8 | 31 (8–42) | 20 (1–33) | 8 (1–22) |
| | | Member | 17 | 27 (22–32) | 22 (14–24) | -- |
| 2 | 5 | Leader | 6 | 31 (21–37) | 20 (10–30) | 11 (2–24) |
| | | Member | 22 | 23 (20–26) | 18 (11–20) | -- |
| 3 | 1 | Leader | 1 | 38 | 32 | 11 |
| | | Member | 3 | 32 (31–32) | 18 (13–22) | -- |
| 4 | 2 | Leader | 2 | 16 (9–22) | 10 (2–18) | 2 (2) |
| | | Member | 5 | 8 (7–9) | 3 (2–3) | -- |
| **Overall** | **14** | **Leaders** | **17** | **28 (8–42)** | **20 (1–33)** | **8 (0–21)** |
| | | **Members** | **47** | **23 (20–30)** | **18 (11–23)** | -- |

**Table 3. Data sources, roles, and data collection methods.**

| Data source | Role in *Mphatlalatsane* | Data collection method |
|---|---|---|
| Advisors | Provided technical quality improvement support and mentoring to teams | Semi-structured interviews |
| Leaders | • Recruited members<br>• Quality improvement induction to members<br>• Managed team activities | Semi-structured individual/group interviews |
| Members | Implemented Plan-Do-Study-Act cycles | |
| Programme documentation | Leaders', advisors, and Project management committee's record of implementation planning and progress | Reviewed documentation |
| Fieldwork journal | n/a | Lead author recorded his fieldwork reflections |
| Project management committee meetings | Coordinated programme implementation | Lead author attended meetings |

We used the Consolidated Framework for Implementation Research (CFIR) to structure our data collection (and analysis, detailed later on). The CFIR is an often-used guide to plan, conduct, and analyse process evaluations of healthcare interventions [37]. The framework comprises 39 'constructs' or implementation factors [40], grouped into five domains: *Intervention characteristics*, *Outer setting*, *Inner setting*, *Individuals*, and *Implementation processes*. We used these domains to focus our data collection: under *Intervention characteristics* we set out to explore how the QI programme was being implemented; we defined *Inner setting* as the hosting facilities and QI teams (see S4 Table: CFIR Domain 3); our *Individuals* of interest, the advisors, and the QI team leaders and members; and we defined *Implementation processes* as all activities undertaken by the individuals to implement and/or manage *Mphatlalatsane*. In 2022 Damschroder and her colleagues added a sixth domain, *Outcomes* [41]. This addition occurred towards the end of our fieldwork, after much progress had been made with data analysis. We therefore did not account for this domain in our study. Other members of the evaluation team evaluated the *Outer setting* domain; thus this is not included in this paper.

**Leader and member individua/group semi-structured interviews and team programme documentation.** Prior to the evaluation, leaders, members, and facility managers were briefed in-person about the evaluation during recruitment in April 2021. Leaders and members were interviewed in a private space at their facilities, at a date and time set by the leader. Ensuring that the data collection did not disrupt service delivery, it was the leader's choice to have them, and the members, interviewed separately or jointly. For the same reason, the leader decided to have members individually interviewed or in groups, which comprised two to four members. The interviews focused on their perceptions and experiences of the QI programme, and the enablers and barriers to team performance (S5 Table). The interviews were conducted at three time points: May 2021 (Timepoint 1), September 2021 (Timepoint 2), and September 2022 (Endpoint). The leaders made their QI documentation available at the second and endpoint data collection and also answered clarification questions at these time points. A total of 71 interviews (Table 4) were conducted, on average 42 minutes long. Apart from replacement leaders in two facilities, all leaders were interviewed at each time point. At the researchers' request to gain variation in member perceptions and experiences, the leaders recruited new members at each time point. In only one facility, some members were interviewed twice.

**Advisor semi-structured interviews and their programme documentation.** The first advisor interview was in February 2020. From February 2020 until November 2022, we conducted 37 interviews using Microsoft Teams (https://www.microsoft.com/en/microsoft-teams/group-chat-software). It was mostly joint interviews, but occasionally individual, if an advisor was unavailable for the joint interview. The interviews focused on: implementation

**Table 4. Leader and Member semi-structured individual/group interviews.**

|  | District 1 (6 facilities) | District 2 (5 facilities) | District 3 (1 facility) | District 4 (2 facilities) | Total |
|---|---|---|---|---|---|
| **Participants** | **25** | **28** | **4** | **7** | **64** |
| Team leader interviews | 13 | 12 | 3 | 4 | **32** |
| Team member individual/group interviews | 15 | 18 | 3 | 3 | **39** |
| **Total interviews** | **28** | **30** | **6** | **7** | **71** |

progress; teams' performance and the influencing factors (S5 Table); and teams' adaptations of the PDSA model. Advisors made their programme documentation available and clarified follow-up questions throughout the study.

We chose the semi-structured interview format because we had a set of topics we wanted to explore (S5 Table), but also wanted participants to share experiences and perceptions that were important to them, even if these were not part of our topics [42]. Our interview guides were firstly informed by the CFIR, and secondly by the QI literature which highlighted issues such as team leadership and methodological adaptations as important aspects of QI programmes. The lead author (WO) piloted the interview guides within the evaluation team prior to data collection. All interviews were conducted in English, audio recorded and sent for professional transcription.

**Fieldwork journal.** WO (a male scientist at the SAMRC who holds a Masters (Research psychology), and with 18 years' qualitative research experience), recruited the participants and collected all the data. He kept a journal, recording his impressions and experiences of the fieldwork.

**Programme management committee documentation and meetings.** The committee included the larger evaluation team in their meetings and shared programme documents and meeting minutes. This provided insight into micro level programme planning and implementation.

## Analysis

The analysis firstly comprised a qualitative assessment of teams' performance done during data collection.

**Team performance assessment.** During the fieldwork and early analyses stages we identified several aspects on which the well-performing teams differed from the less well-performing ones. We distilled these into four criteria. A team had to meet all four criteria to be considered as well-performing:

a. *Leadership*: Evidence of a leader and team structure, as observed during data collection;

b. *Reviving QI*: Evidence, from reviewing programme documentation, of renewed QI activities between August 2020 and March 2021 (this was a challenging period as it followed the cancelling of QI activities between March and July 2020 due to COVID-19);

c. *Leader attitude*: Positive towards QI and held positive views on their team's performance, reported during leader and member interviews and advisor debriefings; and

d. *QI maturity*: The extent the leader and advisor, reported in the interviews and debriefings, thought teams could function without advisor support for the last four implementation months.

For the interview data, WO confirmed transcript accuracy by reviewing them against the recordings. As he read through the programme documentation, information relevant to team

performance was copied into a Word document. This, together with the transcriptions and fieldwork journal were loaded and analysed in Atlast.ti, 8.1 (https://atlasti.com/).

In this study we used framework analysis, with the CFIR as the framework of choice as it focuses on identifying and interpreting how contextual factors and implementation processes influence intervention implementation and effectiveness [37]. With 'contextual factors' and 'implementation processes' being two of the most profound issues to explore in order to understand QI team performance [11, 43], the CFIR are often used in the evaluation of QI programmes [18, 20, 44]. We chose framework analysis as it allows comparative thematic analysis [45]. Key to the method is developing a thematic framework with categories of codes [46, 47]. These categories are informed by what transpires during data immersion and *a priori* issues, amongst others, the study objectives and interview guides [47]. Given the iterative nature of the method, codes and categories were amended during analysis [45]. We followed the steps proposed by Goldsmith et al. [45] to conduct the analysis: after data familiarisation, WO and two senior evaluation team members (XH and TC, henceforth 'analysis team'), used the CFIR to index, chart, and map the dataset as follows:

*Indexing*. After reviewing the definitions of *Domains 2–4*, the analysis team translated it into how it was operationalised in *Mphatlalatsane*, e.g., the *Inner setting* translated into *Facility context and QI team*. We identified the constructs applicable to the evaluation aims and followed the same process of translating it into how it was operationalised in *Mphatlalatsane*, for example, *External agents* translated into *Advisors*.

*Charting*. The analysis team summarised the indexed data onto the selected constructs.

*Mapping and interpretation*. Based on the data analyses, we delineated the life course of a *Mphatlalatsane* QI team. We loosely defined 'life course' as commencing with identifying the leader and their training, through to establishing the team and them developing and implementing their first change idea. We drafted a flow chart (Fig 2) depicting this life course and mapped the CFIR domains and constructs onto the life course stages. We selected examples of good practices for each construct.

In the Results, Section 2 'Applying the Consolidated Framework' reflects our indexing and charting of the CFIR domains and constructs onto our data, and Section 3, reflects our mapping and interpretation. In Section 3 we used the referred 'comparative thematic analysis' to develop the themes presented as the stages over the life course of a QI team. We then carefully reviewed the salient implementation processes per life stage for the well-performing teams, and from these, identified the 'good practices'. The 'good practices' are what we considered as the single most important driver/s of success for each life stage.

## Ethical considerations

In preparing the manuscript, we completed the COREQ checklist [48] to ensure rigor, transparency, and reflexivity with presenting our results. In addition to the University of the Western Cape (South Africa), Human Research Ethics Committee (HREC) approval received 04 February 2020 (BM19/10/16), this study was included in the overall evaluation proposal, which received ethical approval from the South African Medical Research Council's HREC, 30 November 2020 (EC019-11/2019). By registering the study for a PhD with Stellenbosch University (South Africa), we received their HREC approval on 12 October 2021 (S21/05/096). We also received approval for the study from the research committees in the respective provincial health departments: for Eastern Cape, 27 January 2021; Limpopo, 16 March 2021; and Mpumalanga, 8 April 2021.

Participant information documents, including study aims, researcher's details, and consent letters, were shared with participants during recruitment. Their signed, written consent letters, confirming voluntary participation, their right to withdraw at any point, and anonymising the

**Stage 1a: Identifying leaders with quality improvement characteristics**
*Domain 4: Individuals* (Leader)
<u>Constructs</u>
- Knowledge and beliefs about intervention (Intrinsic motivation)
- Other personal characteristics

*Domain 3: Inner setting* (Leader)
<u>Construct</u>
- Learning climate (Fallibility)

**Stage 1b: Appointing qualitative improvement advisors**
*Domain 5: Implementation process*
<u>Construct</u>
- External change agent (Advisor)

**Stage 2: Training leaders**
*Domain 1: Intervention characteristics* (Plan-Do-Study-Act model)
<u>Construct</u>
- Adaptability (Core/peripheral elements)

*Domain 3: Inner setting* (Facility/Existing service team)
<u>Constructs</u>
- Culture (Prevailing norms/values)
- Compatibility (Embed into standard practices)

**Stage 4: Implementation**
*Put into practice*
<u>Constructs</u>
- Adaptability
- Culture
- Compatibility
- Learning climate

*Domain 1: Intervention characteristics* (Plan-Do-Study-Act model)
- <u>Constructs</u>
- Complexity (first change idea)

*Domain 3: Inner setting* (Existing service team)
<u>Construct</u>
- Resources (Facility management)

**Stage 3: Establishing teams**
*Domain 5: Implementation process*
<u>Construct</u>
- Engaging (Core team)

**Fig 2. Consolidated Framework for Implementation Research and quality improvement team life course.**

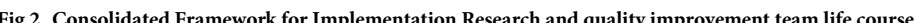

data before publication, were collected before data collection commenced. We further anonymised the participants in this manuscript by replacing the identifying pronouns of 'she/he, hers/his' with 'they/them/theirs'.

## Results

We first summarise the team performance assessments (Table 5) and present the CFIR domains and constructs over the team's life course. We then summarise how these were

**Table 5. Well-performing teams by district and facility type, September 2019 to November 2022.**

| | District 1 (total teams) | District 2 (total teams) | District 3 (total teams) | District 4 (total teams) | Total (total teams) |
|---|---|---|---|---|---|
| **Regional hospital teams** | 1 (1) | 0 (2) | no team | no team | **1** (7) |
| **District hospital teams** | 0 (2) | 0 (1) | no team | 0 (1) | |
| **Community healthcare centre teams** | 2 (2) | 1 (2) | 0 (1) | no team | **3** (5) |
| **Clinic teams** | 1 (1) | 1 (1) | no team | 0 (1) | **2** (3) |
| **Total** | **4** (6) | **2** (6) | **0** (1) | **0** (2) | **6** (15) |

indexed and charted into *Mphatlalatsane* operationalisation, (Section 2: Applying the Consolidated Framework for Implementation Research), followed by Section 3 'Good practices' of mapping and interpreting good practices for the different stages of the life course.

## 1. Qualitative team performance assessment

We rated six of the 15 teams as well-performing, summarised in Table 5 (see S6 Table for the assessment evidence per facility).

## 2. Applying the Consolidated Framework for Implementation Research to the life course stages of a *Mphatlalatsane* quality improvement team

We outlined four stages over the life course of a QI team, starting with *Stage 1*: *Identifying leaders and advisors*. Here we describe the characteristics the data suggested as important when selecting leaders and advisors. While *Stage 2*: *Training leaders*, deals with training, we propose good practices for setting up the team in *Stage 3*: *Establishing teams*. In *Stage 4*: *Implementation*, we offer good practices to the facility management regarding implementation. Though some domains might appear inappropriate to a stage, it was the intention of the CFIR authors to allow its users to adapt the framework to their specific work [37]. In Flowchart 1 we list the CFIR domains and constructs over the life course of the QI teams, with the *Mphatlalatsane* elements in brackets, and in Table 6 provide a definition for each domain [37] (see S7 Table for the indexing and charting of constructs presented in Fig 2).

## 3. Good practices over the life course of a quality improvement team

We present good practices from the six well-performing teams over the teams' life course (Fig 3). The "good practices" are not ranked in order of importance but follow the life course stages.

**Stage 1a: Identifying leaders.** Leaders were the *Inner setting* drivers of well-performing teams:

*Researcher: How much of the team's performance lies with the team leader?*

**Table 6. Consolidated Framework for Implementation Research domains operationalised in Mphatlalatsane.**

| Domain | Definition | Mphatlalatsane operationalisation |
|---|---|---|
| **1: Intervention characteristics** | Intervention description | Plan-Do-Study-Act model |
| **3: Inner setting** | Organisation in which the intervention is implemented | Existing facility service team/s, QI team |
| **4: Individuals** | Individuals who implement the intervention | Leader, Members, Facility management, Advisors |
| **5: Implementation process** | Intervention implementation processes | *Mphatlalatsane* implementation processes |

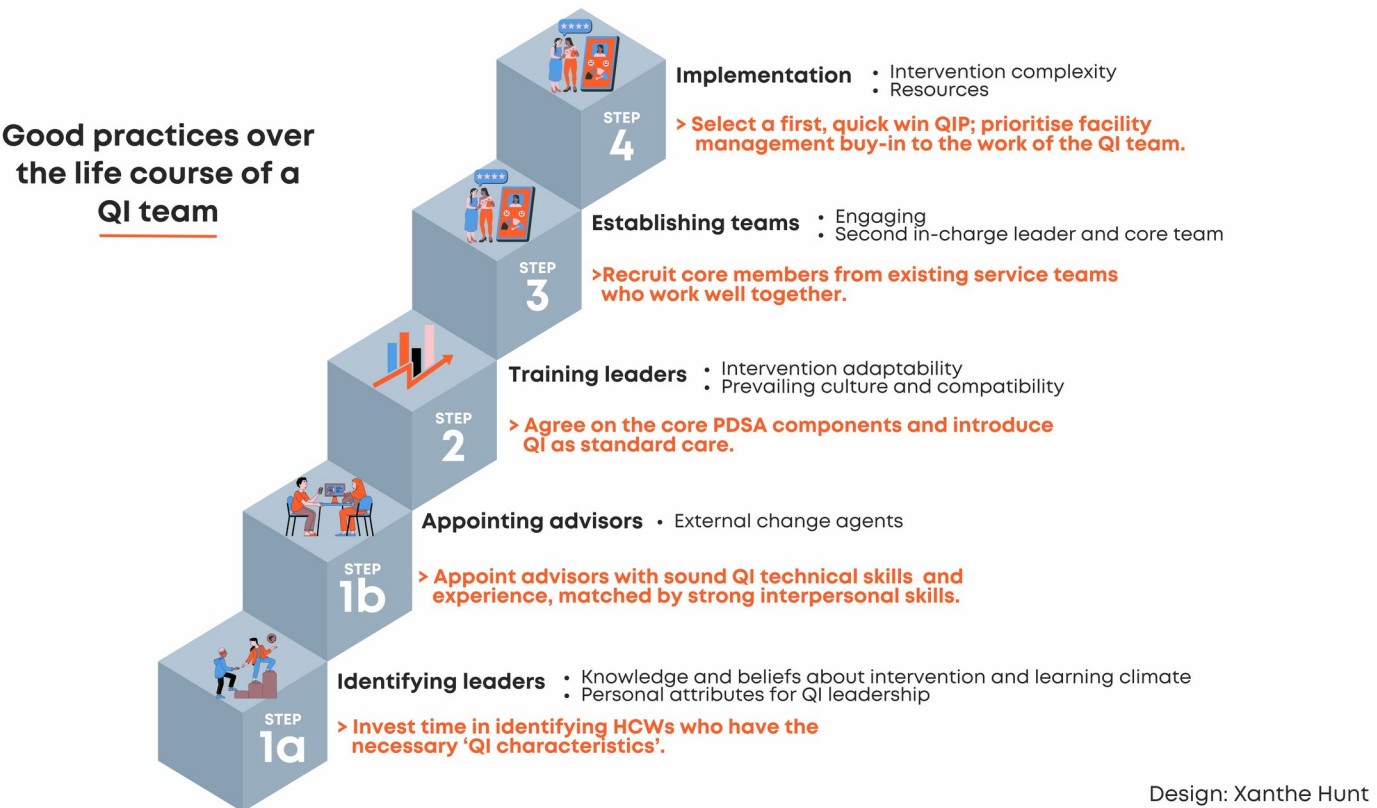

**Fig 3. Good practices over the life course of a quality improvement team.** QI: Quality improvement, HCW: Healthcare worker, QIP: Quality improvement plan (the change idea to solve a service delivery challenge).

*Advisor: I think a hundred percent, it does . . . If the team leader is not doing that, nobody will do it in the facility. (Advisor 4).*

These leaders shared several characteristics, which are detailed in the following subsections: *Leaders are quality improvement enthusiasts.* Foremost is what Damschroder et al referred to as a "positive affective response to the intervention" [37] (p. 9), in this case, QI. A leader of a well-performing team affectively expressed her enthusiasm as follows:

*I'm so in love with quality improvement. (Leader, Well-performing team 1)*

For some, this intrinsic motivation followed from experiencing improvements in the challenges they addressed:

*Researcher: Okay, but then it seems to me that it [two change ideas] ran simultaneously?*

*Leader: Yes, you're right, because after we've seen what we have achieved in Change idea 1, we immediately started on Change idea 2. (Leader, Well-performing team 5)*

For others the methodology that just resonated well with them:

*It's enriching my mind . . . my staff. It's not an extra load, it's a powerful project. (Leader, Well-performing team 3)*

Either way, we observed that for leaders who embraced the methodology, it improved the quality of their services, despite challenges such as staff shortages and keeping night shift members involved in the change ideas.

*Leaders create a learning environment.* QI starts with acknowledging failures in service delivery, and leaders from well-performing teams created a safe environment for members to own their failures. This leader challenged her team to be honest about under-performing on a service indicator, that then became their change idea:

*Okay, from there we then started by looking at ourselves, our performance and we said: 'Okay, in this facility we are not doing well in this so let's start by doing this project'. (Leader, Well-performing team 4)*

Related to owning failures, was leaders' openness to criticism:

*Like we said, [we should be] . . . open to criticism and accepting those criticisms as constructive. Because otherwise we are not going anywhere. (Leader, Well-performing team 1)*

*Leaders are persistent and curious.* Two other 'QI characteristics' that leaders displayed, were firstly, a tenacity to ensure that members implemented the change idea, as described by this leader:

*First thing in the morning, we go to the consulting rooms and then we check. (Leader, Well-performing team 6)*

The second characteristic was curiosity, and having the skills, to understand what routine data tells, evident in the following quote:

*Researcher: How do you know the tool is effective?*

*Leader: Through the graph. We're doing graphs after we have found the results. So, through the graph you can see that you are able to meet the targets. (Leader, Well-performing team 3)*

This curiosity and aptitude to use routine data was absent in most leaders of less well-performing teams, exemplified in this leader whose view was in sharp contrast to the leader above:

*We are clinicians, we're not researchers. We don't know how to do research [referring to the data driven processes of QI] (Leader, Less well-performing team 7)*

## Good Practice 1

Invest time in identifying healthcare workers with quality improvement characteristics to become team leaders: commitment to quality improvement; openness about failures; implementation tenacity; and aptitude to understand and use data.

**Stage 1b: Appointing advisors.**    *Advisors as external change agents.* Another key driver in *Mphatlalatsane* was the advisors. They assisted leaders to master the QI methodology, and their importance was acknowledged in well—and less well-performing teams:

*And now with Advisor X . . . pushing us sometimes to continue with the project, checking us, how far are we, then we decided let us continue. (Member, Less well-performing team 7)*

Though this "pushing" was not needed for leaders from well-performing teams, they appreciated the advisors' mentoring and technical support:

*And then they [advisors] would ask, how are you going to change? Then we told them that we were going to do this and this . . . And then they came and did the support visits and looked at what we were doing. (Leader, well-performing team 4)*

During the COVID-19 pandemic the PMC took a decision to broaden the advisors' scope of work to include meeting the more immediate needs of facilities, for example to support districts with their supply chain management to provide HCWs with personal protective equipment. This trend continued well beyond the height of the pandemic, evident in the advisor reports at PMC meetings of more demands to support non-QI activities. This diluted the advisors' QI support, as described by one advisor:

*We [became] sort of the port of entry, so whatever happens coordination-wise and communication to both the district as well as the province, it has to happen via the QI advisors. It takes a lot away from the actual quality improvement work that we were initially hired for . . . it is now more around health system strengthening. (Advisor 1)*

A notable difference was observed between leaders of less well—and well-performing teams towards the last few months of *Mphatlalatsane*. In the former, leaders reported that they were not ready to function without advisor support, whilst leaders from well-performing teams felt they only need to call on the advisors when they get stuck:

*Yes. it [the team] functions without an advisor, I mean, here [at the facility], but when we are having problems, then I phone Advisor X to help us. (Leader, well-performing team 4)*

*Mphatlalatsane* showed that advisors' interpersonal skills were as important as their technical expertise, and they assisted teams beyond programme activities. This was shown in how they handled the COVID-19 trauma HCWs experienced. A skill-mentoring contact often turned into a debriefing for HCWs about how the pandemic affected them:

*So, what I normally do is when I call, I would ask them about the Covid activities, how are they doing just so that they can see that I'm also not only concerned about the quality improvement work, but I would ask about them, are they having any challenges. . . (Advisor 2)*

## Good Practice 2

Appoint advisors with sound technical skills, matched by strong interpersonal skills, and train them to tailor their support to facilitate team independence.

**Stage 2: Training leaders.**   Our good practices relate to intervention adaptability, the work culture in existing service teams, and intervention compatibility with standard practice.

*Intervention adaptability.* On rare occasion, births were delivered in clinics, and participants reported a monthly average of about 47 deliveries across the CHCs. In contrast, there were on average approximately 400 and 600 births per month, respectively, in the district and tertiary hospitals. The pressures that came with managing high volumes of deliveries was offered by this leader as reason why they failed at times to follow the PDSA model to the letter:

*And the queue is there, when you are still busy there's someone who comes, it might happen with that one if she comes, she's really in labour. Everything must stop. Some, they come at*

*advanced [stage of labour]. When we check, 'Let's go to delivery room!' (Leader, Less well-performing team 2)*

All teams reported that COVID-19 forced them to adapt the PDSA model they were trained on. The unchanged component was keeping it a data driven process, i.e., using data to identify service delivery challenges and assess the effectiveness of change ideas. We considered them the core intervention component. Other adaptations were having long PDSA cycles, and not plotting assessment data on run charts but keeping it in their audit tools. Since these did not affect change idea effectiveness, we considered them peripheral components. This quote refers to the time period when the teams started to revive their QI work after the COVID-19 lockdown:

*All of them [facilities] are still using the methodology, it's just that they are not meeting as regularly as it should be. So, when you get there, instead of them getting you feedback . . . . You'd start by consolidating the process measures [audit tool], for instance. (Advisor 2)*

## Good Practice 3

Agree during QI leader training what the core Plan-Do-Study-Act components are to ensure that the main intervention component/s are implemented.

*Work culture of the existing teams.* Leaders and members from well-performing teams described being proud about how good their existing service teams were pre-*Mphatlalatsane*:

*Researcher: And how do you function as a team (maternity ward staff)? Did it [Mphatlalatsane] change anything on that side?*

*Member: No, our team is doing the same. We are still working very hard like we do. . . we are the best. (Member, Well-performing team 5)*

The opposite was true for less well-performing teams. This leader made it clear that there was a negative work culture in their existing service team. Members objected to more work and the leader offered it as a reason why their QI team was not progressing with its QI activities:

*'No! No!' That's why I said we are not moving anywhere. (Leader, Less well-performing team 2)*

## Good Practice 4

During training, leaders should recruit members from existing service teams known to have a positive work culture and who work well together, and plan to sustain their strengths in the quality improvement team.

*Intervention compatibility with standard practice.* Leaders, both from less well-performing and well-performing teams, had to overcome resistance from members when they introduced *Mphatlalatsane*, because it was perceived as, and often indeed was, more work:

*He or she [leader] is going to experience resistance, especially . . . bringing a change, you're going to experience inertia. . . And then it's [QI] also consuming most of your time and leaving some of your expected activities [routine work] to be done . . . because you are concentrating on getting this [QI] work done. (Leader, Well-performing team 5)*

Yet, many of the change ideas were getting HCWs to follow existing NDoH guidelines and according to this leader, nothing more than what they were expected to do.

*But this [change idea] was something that has been communicated all along [by the Provincial Department of Health], that every childbearing woman, they have to test for pregnancy even if she's coming to the Outpatients department . . . (Leader, Well-performing team 4)*

## Good Practice 5

Quality improvement should be introduced to leaders and members as the standard practice, getting teams back-to-basics, and not as something additional to what they are supposed to do.

**Stage 3: Establishing teams.** Our good practice for team composition involves forming enthusiastic and motivated teams.

*Engaging the right team members.* Well-performing teams had a core team of two to six members, who, together with the leader: identified the service challenges; developed the change ideas; and decided how to assess its outcomes. They recruited ad hoc members from existing staff to implement the change idea. We found that core members were QI enthusiasts who encouraged ad hoc members to participate in change ideas:

*Then we saw that this thing is working. Then we came together and add our input, and we continued motivating other staff [ad hoc members] to continue. (Member, Well-performing team 1)*

The core team included a member who acted as the second in-charge leader to help the leader manage her routine and QI responsibilities, as reported by this leader:

*Ja, what worked for us . . . one of us, she must be in charge [leader] of the project. But she must not be alone. Like what I'm saying, tomorrow I'm off [leader]. [second in-charge leader], she's there . . . she must check what we have been doing. (Leader, Well-performing team 5)*

## Good Practice 6

Establish a core team with members who have influence in the facility and are positive about quality improvement, and appoint a second in-charge leader who can fill in when the leader is absent.

**Stage 4: Implementation.** We found that intervention complexity and facility resources also shaped implementation.

*Intervention complexity.* For some members of less well-performing teams, the methodology did not make sense. An example is a team whose change idea was to improve partogram completeness. Members felt frustrated by reviewing files to establish change idea effectiveness and would rather spend their time with patients. On the other hand, effective change ideas generated enthusiasm amongst members. In one hospital, the team's aim statement was to improve triaging pregnant women on arrival from 0% to 80%. Their change idea included: assigning an admission nurse to triage patients within 10 minutes of arrival; putting a bed in the admission section; and procuring a stamp for the nurse to record the arrival time and triage actions taken. Their triaging improved from 0% to a sustained 80% and above. Members who saw that their change idea reduced complications during delivery were excited about their success:

*Within a week we then saw that this thing was working because when they are triaging, you'll find that this woman is fully dilated while outside, then they took the woman into the labour ward. Then we saw that this thing is working. (Member, Well-performing team 1)*

The MNH issues commonly targeted in the change ideas at the clinics and CHCs were improving anaemia during pregnancy, antenatal first visit booking before 20 weeks, antenatal

TB screening, and antenatal viral load monitoring. In the hospitals the change ideas focused on better management of labour, for example aiming to reduce post-partum haemorrhage, improve partogram completion, and improve infection control in the neonatal wards (see S8 Table for a list of health issues that the *Mphatlalatsane* change ideas targeted).

### Good Practice 7

The first service delivery challenge that the team identifies should be one that can be solved easily with a quick, big-win change idea.

*Facility resources*. In the well-performing hospital, the facility management was not part of the team. The leader narrated how positively they experienced management support: for their triage change idea (described above), the finance manager provided funds to procure the mentioned stamp to audit the change idea, and said it was because he understood their change idea:

> So, *there was no problem with the finance manager buying into that [QI] . . . and then in no time we managed to get the stamp through the Finance manager . . . meaning he understood what impact this would have on the institution. (Leader, Well-performing team 1)*

This was not the case for some leaders of less well-performing teams, who felt in many instances the QI work was dumped onto them:

> This is our CEO, *he's supposed to be on the ball with these things, but he just shoves it off onto us as if it's just a side-line programme. (Leader, Less well-performing team 7)*

### Good Practice 8

Prioritise facility management buy-in to ensure that the team will have what it takes to implement their change ideas.

## Discussion

Though the results corroborate previous research on several implementation processes and contextual factors that impact QI teams [44, 49, 50], it offers a novel 'life course lens' through which we tracked the life stages of QI teams over 39 months. This allowed us to distil six good practices related to implementation processes, and two related to the contextual factors to optimise QI team performance.

### Optimal implementation processes

**Selecting a team leader with QI-specific characteristics.**  The role of the leader is well reported [11, 51, 52]. It refers mostly to general leader traits such as: promoting member ownership of a programme [53]; open communication [11]; and leading with integrity [54]. These were true for *Mphatlalatsane* too and are reported in our past publications [34, 55]. However, we see the leader as the most important driver of team performance and identified four QI-specific traits associated with leaders of well-performing teams. Distinguishing these traits from general leader characteristics will avoid assuming that a HCW with sound general leadership skills will by default be a good QI leader.

The first trait is to embrace QI as a methodology to solve service delivery challenges. Literature describes the importance of being enthusiastic about QI [56, 57], but does not identify it as a leader characteristic. A central component of QI is identifying and correcting service delivery gaps [8, 58]. Doing so without people feeling blamed can be challenging [59]. The

second important QI leader trait is therefore setting an example of owning her/his failings [60], to create "psychological safety" [21] (p.2), where members participate without fear of rejection or blame [21]. This trait was present in leaders of the well-performing teams but absent in leaders of less well-performing teams.

The third characteristic relates to the PDSA model. As found by Katowa-Mukwato et al, where the PDSA model was used to improve services in a female ward, the more complex the intervention, the more challenging it is to adopt the intervention [61]. The PDSA model is such a complex intervention, and as a data driven method [62], equated with conducting research [63, 64]. The leader should, therefore, thirdly, be competent in interpreting data and looking for ways to improve services [65]. In our study, most leaders of less well-performing teams did not display an aptitude to engage with the 'research-like' principles of QI. Service delivery gaps that became entrenched can be hard to change [59, 66]. Successful QI leaders need, fourthly, the tenacity to keep members to task, until the new behaviours become standard practice, a QI-specific leader characteristic for which we did not find evidence in the literature.

**The QI advisor must have sound interpersonal skills.**   The role of the advisors in teams' performance is well documented, but usually about imparting technical skills [67–69]. In an earlier paper we reflected upon their importance, and how, in one instance, the resignation of an advisor, and in another, being less experienced, negatively impacted team performance [34]. What is less reported is the importance of the advisor's interpersonal skills. Their role was like the leaders' in getting members' buy-in and keeping them to task. The leaders were initially PDSA naïve, and it was left to the advisors to get them to use the methodology and encourage them to adopt new behaviours. We recommend that agencies responsible for training QI advisors include a module on interpersonal skills in addition to the technical skills curriculum.

**Managing methodological adaptations.**   The PDSA adaptations we found (detailed in an earlier publication [34]), are not unique to *Mphatlalatsane*. In a systematic review on PDSA fidelity, only 4% of 72 projects followed all four steps [4]. This was confirmed in another review where only 20% of 73 articles used iterative cycles [70]. Both reviews concluded that adaptations could dilute the effectiveness of QI. Our good practice suggestion, that adaptations should be a regulated process, guided by the advisor, is in line with what Reed et al recommend [71]. Though it is not possible to foresee crises such as COVID-19 that can trigger adaptations, it will help to be attentive to enforced changes and how it affects the QI activities.

**Introduce QI as standard practice.**   This study confirms that HCWs often experience QI methodologies as an add-on that distracts them from their daily work [24]. This can be corrected if QI is introduced, and practiced, as nothing more than supporting HCWs to do what they are supposed to do. Teams should be encouraged to use existing standard care guidelines in change ideas [69], and a "well-rooted learning culture . . ." (p. 293) [72] be promoted at the facility [73]. This confirms Kaplan et al's view [43], that routine use of QI may follow when management cultivates a QI culture amongst HCWs.

**Team composition.**   Our final implementation good practice relates to team composition. There is literature on the importance of matching change ideas with the right HCWs [69], and the induction of new team members [15, 74]. Our results add to understanding the effect of team composition on team functioning, by postulating a positive association between having a core team, with a member acting as the second in-charge leader, and team performance. The *Mphatlalatsane* experience showed that it is necessary to have someone in place to lead when the leader is absent to ensure continuity during staff turnover, an issue we did not find reported in the literature. It also illustrated the value of a core team with members who are QI enthusiasts who can influence others [61] to participate in QI activities.

## Optimal contexts

'Context' is the most often reported component that shapes QI programmes [53, 69, 75]. Our evaluation focused on facility level contextual factors and found two less reported contextual factors shaped teams' performance.

**Prevailing culture in existing service teams.** The first contextual factor refers to the work culture in existing service team/s, the "clinical microsystem" [76] (p. 417), from which QI members are recruited. Rowland et al. [76] suggest that differences in function and roles between the existing and QI teams requires attention to understand the reciprocal impact these teams may have on each other. We found that well-performing teams experienced their "clinical micro system" as positive environments, where working well together was the norm. We conclude that the prevailing culture in existing service team/s (as a QI team can comprise members recruited from more than one existing service team [77, 78]), should not be ignored when planning a QI programme. It can be assumed that members from existing team/s where teamwork is the norm may carry that culture into their QI team, and likewise may those from team/s with a negative culture, first need induction to positive teamwork, before commencing with QI activities. We suggest investing time to ensure a positive culture in the existing team/s prior to establishing QI teams. The prevailing culture is also inclusive of the learning climate that leaders foster when they allow and encourage members to learn from their mistakes [21].

**Facility management support.** The second contextual factor relates to when neither the leader nor members are part of the facility management. The importance of management support is illustrated in one study reporting how QI activities were delayed because management did not approve it in good time [57], and conversely, how management approval for staff to attend training, strengthened the QI programme [23]. All the *Mphatlalatsane* hospitals were without direct management participation. The well-performing hospital had management who supported them actively. The other hospital teams, all less well-performing, complained about management's apathy towards their QI work. The absence of management support may not have been the sole reason these teams performed less well but may have played a role in why some teams performed less well. Our good practice infers that QI teams are likely to under-perform when they lack support from facility management [75].

Finally our study confirms the value of conducting a process evaluation of how QI teams are established and function [63, 79].The longitudinal data did not show changes in team performance over time, other than well-performing teams becoming independent of advisor support. The less well-performing teams remained less well-performing. Similarly, the adaptations that teams made early in the programme stayed for the remainder of the implementation period. Since it appears that teams get set in their ways of being enthusiastic, or not enthusiastic, about QI, attention is needed in how programs are set up and managed in its early stages.

## Study strengths and limitations

The main strength of our evaluation was the longitudinal data collection. Though our engagements with teams were only over 16 months, we kept informed about their successes and challenges through the advisors for 39 months. This supported a comprehensive understanding of the enablers and barriers of team functioning and identifying less-reported issues. The facility-based data collection gave insights into HCWs' realities, such as the overcrowding of a maternity ward. These experiences enriched our evaluation. Attending the Programme management committee meetings offered contextual information to understand implementation decisions, particularly during the first six months of COVID-19. Attending the meetings made us aware of why the advisors' work gradually included supporting non-QI activities at facility and district

levels. Our evaluation was also strengthened by being able to continue during COVID-19 through regular advisor debriefings.

The study limitations include that, due to COVID-19 and a delay with ethical approval, we did not conduct baseline interviews following the training and establishing of teams. Furthermore, the leaders recruited the members for the evaluation, with the possibility of them being biased towards members who were positive about *Mphatlalatsane*. Being aware that members could feel pressured to participate because their seniors asked them, we stressed the importance of voluntary member participation to the leaders, and during the member interviews. Additionally, the joint leader and member interviews may have inhibited members from sharing negative views in the leader's presence, and conversely for leaders to report negatively about members. Our endpoint data collection occurred three months before *Mphatlalatsane* ended. Though the continued advisor interviews during those months did not provide new insights, it would have been better if the final leader and member interviews occurred closer to the end of the programme. Lastly, we did not collect quantifiable measures to analyse correlations between team performance and configurations of the factors that impacted performance. Such analyses could have allowed us to rank these factors from most to least impactful.

## Conclusions

Several of the *Mphatlalatsane* QI teams not only survived COVID-19 but performed well. This research proposes a 'life course lens' to set up and manage a QI team. We operationalised and described eight good implementation processes and contextual factors to optimise QI teams' performance. The main drivers for well-performing teams are the leaders' enthusiasm for QI and sustained advisor support, which are important entry points for investment in QI in the future.

## Supporting information

**S1 Table. Mphatlalatsane partnerships.**
(DOCX)

**S2 Table. Socio-economic and health indicators of the selected health districts.**
(DOCX)

**S3 Table. Services delivered at participating health facilities.**
(DOCX)

**S4 Table. Consolidated Framework for Implementation Research, Domain 3.**
(DOCX)

**S5 Table. Interview foci.**
(DOCX)

**S6 Table. Assessment evidence per facility.**
(DOCX)

**S7 Table. Indexing and charting Mphatlalatsane CFIR constructs.**
(DOCX)

**S8 Table. Maternal and neonatal health issues addressed through change ideas.**
(DOCX)

## Acknowledgments

Our sincerest appreciation to the five QI advisors for their enthusiastic participation. We are indebted to the frontline healthcare workers who generously shared their time and insights during the evaluation. We also want to acknowledge the provincial and district management staff who supported the evaluation and facilitated our site entry. The NDoH was the custodian of the *Mphatlalatsane* programme, and the Clinton Health Access Initiative acted as the *Mphatlalatsane* implementation secretariat.

## Author Contributions

**Conceptualization:** Willem Odendaal, Mark Tomlinson, Ameena Goga, Terusha Chetty, Xanthe Hunt.

**Data curation:** Willem Odendaal, Terusha Chetty, Xanthe Hunt.

**Formal analysis:** Willem Odendaal, Terusha Chetty, Xanthe Hunt.

**Funding acquisition:** Ameena Goga.

**Investigation:** Willem Odendaal.

**Methodology:** Willem Odendaal, Mark Tomlinson, Terusha Chetty, Xanthe Hunt.

**Project administration:** Willem Odendaal, Ameena Goga, Terusha Chetty, Xanthe Hunt.

**Resources:** Willem Odendaal.

**Software:** Willem Odendaal.

**Supervision:** Mark Tomlinson, Terusha Chetty, Xanthe Hunt.

**Validation:** Willem Odendaal, Terusha Chetty, Xanthe Hunt.

**Visualization:** Willem Odendaal, Mark Tomlinson, Ameena Goga, Yages Singh, Shuaib Kauchali, Carol Marshall, Yogan Pillay, Manala Makua, Terusha Chetty, Xanthe Hunt.

**Writing – original draft:** Willem Odendaal.

**Writing – review & editing:** Mark Tomlinson, Ameena Goga, Yages Singh, Shuaib Kauchali, Yogan Pillay, Manala Makua, Terusha Chetty, Xanthe Hunt.

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
