## [Decision Letter · Decision Letter 0]

22 May 2024

PONE-D-23-40800Good practices to optimise the performance of maternal and neonatal quality improvement teams: results from a longitudinal qualitative evaluation in South Africa, before, and during COVID-19PLOS ONE

Dear Dr. Odendaal,

Thank you for submitting your manuscript to PLOS ONE. After careful consideration, we feel that it has merit but does not fully meet PLOS ONE’s publication criteria as it currently stands. Therefore, we invite you to submit a revised version of the manuscript that addresses the points raised during the review process. Although a bit divided, the reviewers have made extensive comments for your consideration that will no doubt strengthen the paper. I look forward to reading the next submission.

We look forward to receiving your revised manuscript.

Kind regards,

Tara Tancred, PhD

Academic Editor

PLOS ONE

Journal Requirements:

4. In the online submission form, you indicated that [The full data set cannot be shared publicly because of the regulations imposed by the two respective Human Research Ethics Committee's (HRECs) who approved the study. Access to the data can be requested from the respective HRECs via the corresponding author, for researchers who meet the criteria for access to confidential data.]. 

Reviewers' comments:

Reviewer's Responses to Questions

**Comments to the Author**

1. Is the manuscript technically sound, and do the data support the conclusions?

Reviewer #1: Yes

Reviewer #2: Partly

Reviewer #3: Partly

2. Has the statistical analysis been performed appropriately and rigorously? 

Reviewer #1: N/A

Reviewer #2: Yes

Reviewer #3: N/A

3. Have the authors made all data underlying the findings in their manuscript fully available?

Reviewer #1: No

Reviewer #2: Yes

Reviewer #3: Yes

4. Is the manuscript presented in an intelligible fashion and written in standard English?

Reviewer #1: Yes

Reviewer #2: No

Reviewer #3: Yes

5. Review Comments to the Author

Reviewer #1: Dear Authors,

Thank you for submitting a high quality article, tackling an important, relevant and current issue, namely, quality improvement in maternal and neonatal health services. Please check and make the below suggested minor corrections:

1. Abstract: Please revise the statement in lines 39-41 for better readability/understanding: "To identify good

practices to improve quality improvement teams’ performance, by identifying optimal facility-level

contexts and implementation processes."

2. Findings: The findings are insightful. However, as the study is framed from the outset as a longitudinal qualitative evaluation, perhaps it might be helpful to go beyond a description of the time sequence in the background/introduction/data collection sections to bring out this longitudinal element in the findings as well. It is great that the time sequence for recruitment, trainings, implementation is brought out in detail but then how QI processes/outcomes changed (or did not change) with time isn't very clear in the findings, dimming the study's distinctive longitudinal aspect. To clarify, are any of the good practices or contextual factors shaped by time lapse?

Thanks

Reviewer #2: This manuscript describes an important project with key lessons about improving matrnal and neonatal healthcare services in diverse organizations in low and middle-income countries, Overall, the work merits publication. However, the manuscript raised several questions, and the language wasn't always entirely clear.

In the Introduction, it isn't clear if the first sentence applies to all countries or to low-and-middle income countries only. The definition offered of healthcare quality is limited. Perhaps the definition from the Institute of Medicine (US) Committee on Quality of Health Care in America in the publication, Crossing the Quality Chasm: A New Health System for the 21st Century, would better suit the purpose of this work because the IoM definition specifically includes consideration of implementation of effective evidence-based practice. In line 60, the word models is used twice.

Reference is made to the Consolidated Framework for Implementation Research as the basis of this evaluation. The version cited of the CFIR is 2009. If the updated version from 2022 had been used, would it have made any difference to the evaluation and the findings?

Under Setting, the distribution of facilities involved in the work seems appropriate for the delivery of the services in South Africa. However, it would be helpful to know a little more about the institutional maternal mortality ratio, for example, is it applied nationally and what does it mean, and if performance on this ratio was a factor in the selection of the participant facilities.

On page 7, it is unclear what a "partner" is and how they were identified, designated or selected.

On page 8, under Leaders, line 164 refers to both midwives and nurses. This is a little confusing. Is the experience cited all in midwifery, or in the nursing profession, which may not have been specializing in midwifery.

On page 10, under Advisors, it wasn't clear why there were so many resignations from this role. Did the resignations have anything to do with the advisory role and barriers or difficulties faced by the advisors who resigned?

Under Team Performance Assessment, page 13, beginning with line 242: What was the basis for the criteria selected to define well-performing teams, and how were decisions made that teams met or didn't meet the criteria, that is, the operational definitions of the criteria.

Page 16 and following, section 2, see prevous comment on the version of the CFIR used for this work, and if the updated version would have changed the approach to analysis. It was useful to see how the CFIR was used, but the question is just anout the version of CFIR and would it have made any difference if the current version of the CFIR had been used.

Page 25, Intervention compatibility with standard care. The point made earlier about the definition of quality is relevant to the concept of implementing "standard care". The text in this section made an extremely critical point, and the good practice point derived from the text is very valuable for many teams trying to achieve substantial improvements in their services.

Overall, the good practice points drawn out in the manuscript are helpful practical points for others.

The content on Implementation processes and Context also is very helpful for those who might be taking on a similar project in similar circumstances. The study strengths and limitations were worth noting.

An overall concern about the manuscript are the slips in English grammar. A few examples: In the Abstract, the second statement (lines 39 to 41) is not a complete sentence. A consistent issue is the lack of subject and verb agreement, for example, page 4, line 84, the verb should be is not are; line 89, the verb should be has not have. Also, commas were inserted when they were not appropriate.

Overall, the manuscript might be stronger if it were shorter. The authors could consider if some of the content could be summarized or shortened.

Reviewer #3: Summary

This paper discusses its findings into how to produce effective QI within healthcare using the Plan-Do-Study-Act (PDSA) method. They discuss the characteristics of what makes a good QI leader, advisors, and the need for supporting team members.

Strengths:

- The paper explores the users’ perceptions on what worked and didn’t work when implementing QI practices into maternal and neonatal healthcare services.

- The paper provides some interesting ideas that need further exploration

- For the most part, the paper is well written and easy to follow

However, there are some areas that if improved upon would make for a better submission:

Abstract:

- The abstract does not clearly detail what this study is about, does not clearly identify the need of this study, and provides unnecessary detail in the abstract (i.e., analysed using Altas.ti). Additionally, it didn’t clearly define what the core contribution of this work is. It appears as though the contribution is: How to establish effective QI teams within healthcare settings to improve health outcomes: a case study.

Introduction:

- The paper starts off by discussing the use of PDSA in the context of maternal and neonatal healthcare services. However, the body of the results focus on generalised outcomes that the authors already recognise as previously discussed, as most of the discussion was stating that they ‘confirm’ what others have already established. I was expecting the paper to more heavily focussed on how PDSA was used to identify how to improve maternal and neonatal healthcare, what the main obstacles are towards implementation within the context of this work, and suggestions about how this could be improved in future studies.

Methodology

- It isn’t clear as to why PDSA was used rather than six-sigma or lean, but I also expected to have a more indepth discussion about the use of PDSA in the methodology and how it was used in the context of this study.

- What type of interviews were conducted? Where they semi-structured or structured and why?

- What is framework analysis, why did you use this method? Being that you had interview data, why not use deductive thematic analysis?

Results

- Table 6 and the paragraph below it give the exact same information, please decide which method you think best portrays this information.

- You use pronouns when discussing your participants instead of anonymising them with they/them/their

- I found that most of the supporting qualitative data was from the well-performing teams. It would have been interesting to delve deeper into what impacted some teams more than others and how could these be better alleviated?

- Your good practices boxes where interesting, but seemed very ‘well known’ for the most part. Additionally, for GP2, you mention that they should appoint advisors with sound technical skills; who would do this? How can they ensure that have the skills required to assist the team?

- GP4 discusses recruiting ‘members from existing service teams known to work well together’, however, I disagree with this statement. Having a diverse team can lead to better improvements as differing ideas and approaches can lead to more successful changes, it is more about having a team that is willing to try different approaches and learn from failures.

- GP8, I would extend this to, everyone needs to be on board in order to reap the most benefits from incorporating QI training.

Discussion

- In the discussion you provide the implantation processes. I think it would have been easier to digest had the key points been in a table and then expanded upon in the text.

- A lot of the discussion, as mentioned previously, uses phrasing such as ‘this study confirms’. Although it is good to confirm others research, it is difficult to identify what is a novel contribution and what is already established knowledge.

Minor issues:

- There are minor grammatical errors throughout the paper

- Some of the titles are very long

- Sometimes for the quotes it just says R instead of Researcher

I believe this paper has some interesting concepts that can be explored further, but I really wanted to see some of the outcomes of the study in regards to the scenario it was used in and not just insights into carrying out effective QI programs. I also understand the effect that COVID-19 had on this work and applaud the authors for being able to continue their research despite the difficulties. Currently, I do not think this paper has enough of a significant contribution to move forward and the narrative needs major adjustments. However, there is still interesting aspects of this work that could be impactful if restructured.

Some useful papers that I didn’t see in the citation list:

Implementing Evidence Based Practice nursing using the PDSA model: Process, lessons and implications, Katowa-Mukwato Patricia et al. (2021)

Hill, J.E., Stephani, AM., Sapple, P. et al. The effectiveness of continuous quality improvement for developing professional practice and improving health care outcomes: a systematic review. Implementation Sci 15, 23 (2020). https://doi.org/10.1186/s13012-020-0975-2

Damschroder, L.J., Yankey, N.R., Robinson, C.H. et al. The LEAP Program: Quality Improvement Training to Address Team Readiness Gaps Identified by Implementation Science Findings. J GEN INTERN MED 36, 288–295 (2021). https://doi.org/10.1007/s11606-020-06133-1

6. PLOS authors have the option to publish the peer review history of their article (what does this mean?). If published, this will include your full peer review and any attached files.

Reviewer #1: No

Reviewer #2: No

Reviewer #3: No

---

## [Author Response · Author response to Decision Letter 0]

4 Aug 2024

Thank you for the constructive comments which lifted the quality of the manuscript. Kindly review our rebuttal in the Responses document and the corresponding changes in the revised manuscript.

---

## [Decision Letter · Decision Letter 1]

9 Sep 2024

PONE-D-23-40800R1Good practices to optimise the performance of maternal and neonatal quality improvement teams: results from a longitudinal qualitative evaluation in South Africa, before, and during COVID-19PLOS ONE

Dear Dr. Odendaal,

Thank you for submitting your manuscript to PLOS ONE. After careful consideration, we feel that it has merit but does not fully meet PLOS ONE’s publication criteria as it currently stands. Therefore, we invite you to submit a revised version of the manuscript that addresses the points raised during the review process.

Many thanks to the authors for this revised manuscript. In addition to the grammatical changes requested by Reviewer 2, which I agree need addressing, please revise the following. Critically, there is still some clarity needed in the analytical approach taken as it relates to the different sections of the results:

-Bottom of page 5 “*The evaluation and evaluation team were independent…*” reads a bit oddly. Perhaps just “The evaluation team was…”

-Very minor, but for the numbered list on page six, please replace the commas at the end of 1 and 2 with semi-colons

-I’m a bit surprised (given that this was supported by IHI) that it wasn’t the collaborative model for QI with learning sessions being indicated. Not necessarily an issue, just different language than what IHI usually adopts.

-You refer to advisors and note “henceforth ‘advisors’” on pg. 5, so you don’t need to repeat on pg. 8.

-On page 8, please refer to “districts” when you’re referring to more than one (e.g. “districts 3 and 4” vs “District 3 and 4”).

-It’s a bit confusing on pg. 9 that you note only one facility declined due to COVID interruptions, but then note another team in District 4 declined (and there is one team per facility). To this sentence please revise to “…their own participation was sufficient” to add clarity. That being said, in line 168 you note 15 teams included from the consenting facilities—but if there were only 14 consenting facilities (with 15 teams), and if one team declined, then should this be 14? You might want to then clarify that there were 14 facilities, 17 leaders (1 per facility plus replacements due to turnover) and 14 teams (14/15 facility teams, one facility with two teams) that participated. Otherwise, please clarify

-Please swap the semi-colons in line 175 with commas.

-In line 187, please change “initially” to “initial” (or change to “three initially-appointed…”) and add “being” before “replaced”

-Please remove the comma from line 209.

-In line 219 the hyphen is not necessary.

-Please add a colon after “focused on” so that the list is introduced.

-Please provide a bit more information about how the interview guides were developed/what they included. Further, (if you can) please make the links between data collection and analysis. For instance, as you analysed against CFIR (which I imagine was quite the undertaking…), I would have expected to see CFIR domains being covered in data collection to facilitate this, but the information that’s present suggests that was not the case. Otherwise it’s rather like shoehorning your data into a framework that may not actually work. Basically, there may be a disconnect between data collection and analysis. If so, consider adding this as a limitation to your discussion.

-Please introduce what you are listing (presumably the steps by Goldsmith et al) from line 286.

-Please define how you are using “life course” within the confines of this research as it’s not quite described.

-You note that you use a framework analysis, but your results for sections 2 and 3 (though both indeed appearing to be a framework analysis) were derived differently, for instance, with CFIR only being used in section 2 (if it was used for section 3, that is not evident and would need explaining) versus being more thematic in section 3 with the inductively derived themes (the bolded “subsections” in the results, to me, are your themes and I would call them as such). It’s not quite clear how you arrived at your analysis for section 3. Also related to your analysis/results, I really like the “good practice #...” at the end of each results section, but it needs to be clearer from your analysis how you arrived at these. Can you please revise so that wording of the analytical process/results better align?

-Further to this, please add a line to explain why you chose to structure your analysis around CFIR for section 2 versus the many other analytical approaches that could have been taken.

-It’s not clear how the team performance assessment analysis was taken into consideration in the later analysis. Please make this more explicit. Critically, it’s also not quite clear how you populated the team performance assessment—what exactly were those preliminary sources of data? And how did you ensure this wasn’t a totally subjective exercise?

-Please rename “Flowchart 1” to “Figure 1” and swap with Table 6 so that the table comes before the figure as the table facilitates figure interpretation.

-Please add a legend for Figure 2 (e.g. define HCWs, QI, and QIP)

-In line 440, please remove the “a” before “curiosity”

-In line 447 add “use” (or something to that effect) after “data”

-For good practice 2, I wonder if it’s more “..and train them to tailor their support to facilitate team independence…” or something along those lines?

-Please add a comma after “respectively” in line 500 and remove the comma from 501.

-In 514 please replace “it” with “them”

-Please remove the underscore from line 525

-Please change “team” to “teams” in line 526

-Please clarify lines 533/34. It’s not clear what “it” in line 534 is referring to.

-It’s not clear for me how good practice 4 reflects the preceding results. Consider expanding these slightly to make the link clear.

-Please add “both” before “less” in line 542

-Good practice 5: Standards of care usually refer to clinical standards relating to treatment, etc. I think you may mean “standard practice”. Please reflect across Figure 2 and also in line 684.

-Please change the comma in line 565 with a colon and add a colon after “included” in line 593. In general, whenever you have items in a list separated by semi-colons, they need to be introduced with a colon.

-In line 565/66, I think the steps should be plural (e.g. “the change idea” should be “change ideas”).

-In line 574/583/695 please write “2^nd^” as “second” and “1^st^” as “first” in line 603

-Please remove the hyphen from 575 and the comma in line 596.

-Sorry for the semantics, but in 589, the change idea wouldn’t (or shouldn’t) be “partogram completeness” but what they proposed to put in place to support more completion of partograms. Likewise, in 604/5, reducing PPH, partogram completion, and reducing neonatal sepsis aren’t change ideas but aims. Please revise.

-In line 606 you refer to the S7 file as a list of change ideas, but these are just problems to which change ideas were applied. Please revise.

-Please de-capitalise “triage” in line 613

-For all indented and italicised quotations, please remove the quotation marks around them.

-Please write it as “QI-specific” in line 640

-Please add a colon after “as” in line 641

-Please change “leader” to “leadership” in line 646

-Please remove the comma from line 651

-Please replace “should thirdly,” with “should, therefore, thirdly,” in line 659

-Please add a comma after “documented” in line 667 and delete the second one in line 669

-Throughout, where you refer to “contexts” I think “contextual factors” (e.g. line 702) reads more clearly.

-You refer to “prevailing culture in existing service teams” as being a “lesser reported” issue, but I’d argue there’s a substantive literature about this, although it’s often referred to as a “culture of QI” or “improvement” or “teamworking” (juxtaposed to a “culture of blame”), which I would recommend you cite here.

-Please replace “timely” with something like, “within good time” in line 719

-In line 720, should “strengthen” be “strengthened”?

-In line 724 please revise to “…well, but are likely…”

-Change “infer” to “infers” in line 725

-Consider deleting 728–735. I think it’s clear that the good practices reflect what was needed in less well-performing teams and present in the well-performing ones.

-Lines 737–739 are confusing. Throughout the manuscript, manuals are not raised, nor is implementation research. I would delete this after the [60,80] reference and then move to the comment about longitudinal data.

-You don’t refer to stationery (line 751) elsewhere. Please remove.

-You note the value of the program management committee meetings, but it’s not evident how insights from these meetings are reflected in your results. Please incorporate within your results.

-Please delete the hyphen in lines 762 and 765

-In your limitations, please acknowledge that the leader will also likely not speak freely about the “performance” of the members/the team. Further, the leader who declined their team’s participation—this is interesting and I wonder if it’s less to do with their time and more to do with not wanting them to have the opportunity to speak.

-As Reviewer 2 noted, please remove the spaces, throughout the manuscript, around hyphens.

After these changes are incorporated I would be happy to see this manuscript accepted.

We look forward to receiving your revised manuscript.

Kind regards,

Tara Tancred, PhD

Academic Editor

PLOS ONE

Journal Requirements:

Reviewers' comments:

Reviewer's Responses to Questions

**Comments to the Author**

1. If the authors have adequately addressed your comments raised in a previous round of review and you feel that this manuscript is now acceptable for publication, you may indicate that here to bypass the “Comments to the Author” section, enter your conflict of interest statement in the “Confidential to Editor” section, and submit your "Accept" recommendation.

Reviewer #1: All comments have been addressed

Reviewer #2: All comments have been addressed

2. Is the manuscript technically sound, and do the data support the conclusions?

Reviewer #1: Yes

Reviewer #2: Yes

3. Has the statistical analysis been performed appropriately and rigorously? 

Reviewer #1: N/A

Reviewer #2: Yes

4. Have the authors made all data underlying the findings in their manuscript fully available?

Reviewer #1: No

Reviewer #2: Yes

5. Is the manuscript presented in an intelligible fashion and written in standard English?

Reviewer #1: Yes

Reviewer #2: No

6. Review Comments to the Author

Reviewer #1: The authors have addressed, to the best of my knowledge, all comments. Where they disagree with suggested changes (peer review comments), they have outlined their reasons, which I respect and take note of.

Reviewer #2: This manuscript has been substantially improved by the authors' revisions. The authors' analysis and good practice points are extremely helpful to people working in the field of quality improvement.

With awareness that manuscripts are not copyedited by the journal, there are just a few grammar and punctuation issues that should be modified as follows:

Pages 5 and 6: Last sentence, the verbs referring to "team" do not agree. I think "were" should be "was" in the last line of Page 5.

Page 8, line 142: I think the authors mean "IHI's PDSA training" not "IHS's"

Page 8, line 151: Reference to "trainings" is awkward. It would normally be "training sessions" or "training workshops" or equivalent.

When the authors use hyphens (-), the hyphen may not be correct and should be deleted (Page 13, lines 219-220 ("member perceptions and experiences,"), Page 15, line 262 ("leader and member")) or there is often a space between the text and the hyphen, which is not normal practice (Page 8, line 157 ("day-to-day"), Page 33, line 661 ('research-like"))

Some commas should be deleted (Page 26, line 501 "deliveries was offered", Page 30, line 596 "delivery were excited", Page 33, line 669 "reported is the importance")

There is a small word missing before "reason" on Page 26, line 501: "a reason"

7. PLOS authors have the option to publish the peer review history of their article (what does this mean?). If published, this will include your full peer review and any attached files.

Reviewer #1: No

Reviewer #2: **Yes: **Nancy Dixon

---

## [Author Response · Author response to Decision Letter 1]

28 Oct 2024

Kindly find our point-by-point response to the reviewer comments. Please convey our appreciation to the reviewer for the constructive comments.

---

## [Editor Report · Decision Letter 2]

5 Nov 2024

Good practices to optimise the performance of maternal and neonatal quality improvement teams: results from a longitudinal qualitative evaluation in South Africa, before, and during COVID-19

PONE-D-23-40800R2

Dear Dr. Odendaal,

We’re pleased to inform you that your manuscript has been judged scientifically suitable for publication and will be formally accepted for publication once it meets all outstanding technical requirements.

Kind regards,

Tara Tancred, PhD

Academic Editor

PLOS ONE

Additional Editor Comments (optional):

Many thanks to the authors for this revised manuscript. This is reading with a lot more clarity and I am happy to accept.
---

## [Editor Report · Acceptance letter]

7 Nov 2024

PONE-D-23-40800R2 

PLOS ONE

Dear Dr. Odendaal, 

I'm pleased to inform you that your manuscript has been deemed suitable for publication in PLOS ONE. Congratulations! Your manuscript is now being handed over to our production team.

Kind regards, 

on behalf of

Dr. Tara Tancred 

Academic Editor

PLOS ONE